# Anticancer Activity of Snake Venom Against Breast Cancer: A Scoping Review

**DOI:** 10.3390/toxins17100477

**Published:** 2025-09-25

**Authors:** Eun-Jin Kim, Jang-Kyung Park, Soo-Hyun Sung, Hyun-Kyung Sung

**Affiliations:** 1Department of Pediatrics of Korean Medicine, Korean Medicine Hospital, Dongguk University Bundang Medical Center, Seongnam 13601, Republic of Korea; utopialimpid@naver.com; 2Department of Korean Medicine Obstetrics and Gynecology, School of Korean Medicine, Pusan National University, Yangsan 50612, Republic of Korea; vivat314@pusan.ac.kr; 3Department of Policy Development, National Institute of Korean Medicine Development, Seoul 04554, Republic of Korea; 4Department of Education, College of Korean Medicine, Dongguk University, Gyeongju 38066, Republic of Korea

**Keywords:** bioactive peptides, venom-derived toxins, anti-cancer mechanism, drug delivery systems

## Abstract

Breast cancer remains a leading cause of cancer-related mortality worldwide, necessitating innovative therapeutic approaches. This scoping review summarizes experimental evidence on the anticancer activity of snake venom and its bioactive components against breast cancer, drawing from a variety of in vitro and in vivo studies. Aimed at critically evaluating the therapeutic potential and underlying mechanisms, this review consolidates findings on venoms from multiple snake species, including both crude preparations and purified proteins or peptides, revealing a diversity of mechanisms of action. Reported effects include induction of apoptosis, generation of reactive oxygen species, disruption of cell membrane integrity, inhibition of cell proliferation and metastasis, and modulation of oncogenic signaling pathways. In vivo findings further indicate tumor growth inhibition and, in some cases, enhanced efficacy when venom-based agents are combined with nanoparticle delivery systems or conventional anticancer drugs. However, a significant proportion of evidence is limited to in vitro studies, with substantial heterogeneity in venom sources, extraction methods, dosages, and cancer models, which constrains generalizability. There is also a lack of systematic data on long-term toxicity, immunogenicity, off-target effects, pharmacokinetics, and formulation challenges. Taken together, these findings highlight snake venom-derived compounds as promising multi-targeted anticancer agents but underscore the urgent need for standardized formulations, rigorous preclinical safety assessments, and translational research to bridge the gap to clinical application. Future investigations should aim to isolate novel venom-derived compounds, refine delivery strategies, and undertake rigorous preclinical safety and pharmacokinetic studies—ultimately moving toward early-phase clinical evaluation to bridge the translational gap and assess the therapeutic potential of these agents.

## 1. Introduction

Breast cancer accounts for roughly 13% of all cancer diagnoses and remains the most common malignancy among women, with an estimated mortality rate of approximately 15% [1,2]. Although early detection methods and treatment modalities—such as chemotherapy, radiotherapy, hormone therapy, and targeted agents—have improved considerably over recent decades, persistent challenges including drug resistance, severe side effects, and high recurrence rates continue to limit patient outcomes [1,2,3]. Current therapeutic approaches also vary by molecular subtype. Hormone receptor-positive breast cancers are commonly treated with endocrine therapies, while human epidermal growth factor receptor 2 (HER2) positive tumors benefit from HER2-targeted agents [4]. However, triple-negative breast cancer, which lacks these targets, remains difficult to treat and is associated with poorer prognosis [5,6]. The need for novel, effective treatments is particularly critical for these aggressive subtypes and patients exhibiting resistance to existing therapies.

Natural toxins and bioactive molecules have long served as important leads for drug discovery. Among these, snake venom has emerged as a particularly intriguing source due to its complex biochemical makeup and potent biological properties [7,8,9]. Venom contains a diverse array of proteins, peptides, and enzymes—such as phospholipase A_2_, metalloproteinases, disintegrins, L-amino acid oxidases, and C-type lectins [7,8,9]. Research spanning several decades has shown that whole venom and isolated components can selectively induce cytotoxic effects in many cancer cell lines, including breast cancer cells, while sparing most normal cells [7,8,9,10].

Multiple mechanisms are implicated in the anticancer action of snake venom: triggering apoptosis through mitochondrial membrane depolarization and caspase activation [11]; generating reactive oxygen species (ROS) and oxidative stress [9,11]; compromising cell membrane integrity and inhibiting proliferation [7,10]; restraining metastatic spread by reducing cell migration, invasion, and epithelial–mesenchymal transition (EMT) [11]; and modulating oncogenic signaling cascades such as PI3K/Akt, MAPK, and NF-κB [9,12]. In vivo studies in animal models have reinforced these findings, showing that certain venom-derived compounds can suppress tumor growth and, when paired with nanotechnology-based delivery systems or conventional chemotherapy, may enhance therapeutic efficacy [10,13]. Recent advances highlight that the integration of venom components with novel delivery platforms, such as nanoparticles and liposomes, not only improves targeting and bioavailability but also reduces systemic toxicity, thus holding promise for overcoming current limitations of conventional anticancer therapies [8,14]. Despite these promising developments, comprehensive investigations specifically focusing on breast cancer remain limited, underscoring the need for a systematic synthesis of preclinical evidence to elucidate therapeutic potential and guide future translational research [8,15].

While previous reviews have summarized the anticancer properties and molecular mechanisms of snake venom and its components, most notably across various cancer types—including the roles of apoptosis induction, cell cycle arrest, and anti-metastatic effects—these surveys have primarily provided broad overviews and lacked a focused synthesis for breast cancer specifically [16,17]. In particular, prior studies have offered limited discussion on the subtype-dependent responses of breast cancer cells, contemporary trends in nanotechnology-based delivery, and the translational challenges unique to breast cancer models. In contrast, the present scoping review provides a comprehensive and up-to-date analysis with several novel insights: (i) it systematically dissects the efficacy of snake venom and its bioactive constituents across breast cancer molecular subtypes, (ii) incorporates recent experimental advances in nanoparticle-mediated and hybrid delivery platforms that enhance tumor selectivity and safety, and (iii) critically appraises the translational barriers—such as pharmacokinetics, immunogenicity, and formulation standardization—facing the clinical development of venom-derived agents in breast cancer. These perspectives bridge important knowledge gaps not addressed in prior overviews, supporting future precision oncology approaches and translational research directions in this field

## 2. Results

### 2.1. Study Description

The search identified a total of 243 studies. After screening titles and abstracts, 205 articles unrelated to snake venom and breast cancer were excluded. Full texts of the remaining 38 studies were assessed, and 26 studies [18,19,20,21,22,23,24,25,26,27,28,29,30,31,32,33,34,35,36,37,38,39,40,41,42,43] were ultimately included in our review. These studies were conducted across 15 countries. Nations with more than two studies included Brazil (*n* = 7), Saudi Arabia (*n* = 3), Egypt (*n* = 2), and India (*n* = 2), totaling five countries. The study selection process is illustrated in Figure 1.

### 2.2. Analysis of Experimental Methods

A total of 26 studies [18,19,20,21,22,23,24,25,26,27,28,29,30,31,32,33,34,35,36,37,38,39,40,41,42,43] were included in this scoping review. Among them, 18 studies conducted in vitro experiments using breast cancer cell lines. One study [26] combined in vitro and ex vivo analyses, another [37] incorporated in vitro, ex vivo, and in vivo approaches. Additionally, four studies [38,39,40,43] used in vivo models and two studies [41,42] used ex vivo models. Overall, in vitro approaches were the most prevalent, featured in 21 of the 26 studies. In vivo methods were reported in four studies, utilizing xenograft breast cancer models [38,39,40] or chick chorioallantoic membrane (CAM) assays [37]. Ex vivo models were applied in four studies using human breast cancer tissue samples [41,42] and aortic ring fragments from BALB/c mice [26,37].

Most studies used breast cancer tumor cell lines such as MDA-MB-231 and MCF-7, while T-47D [29] and SK-BR-3 [20] were each employed in one study. MDA-MB-231 represents a triple-negative breast cancer cell line, characterized by the absence of estrogen receptor (ER), progesterone receptor (PR), and HER2 [44]. MCF-7 is a luminal A subtype breast cancer cell line that is positive for both ER and PR, and it is widely used in research focused on ER-positive breast cancer [45]. T-47D, another luminal A subtype cell line also expressing ER and PR, is distinguished by its high sensitivity to progesterone and its ability to exhibit progesterone-specific effects independent of estrogen regulation [46]. SK-BR-3 is a breast cancer cell line characterized by high expression of HER2 with negative for both ER and PR [47]. And two studies [26,37] used human umbilical vein endothelial cells (HUVECs) to perform angiogenesis assays.

Most of the reviewed studies focused on evaluating the anticancer effects of snake venom or its bioactive components in breast cancer cells. Erlista et al. (2023) [24] conducted a comprehensive proteomic analysis of *Naja kaouthia* venom and identified potent anticancer peptides derived from its trypsin hydrolysate in MCF-7 cells. Meanwhile, Kisaki et al. (2021) [30] investigated the proteomic changes in MCF-7 and MDA-MB-231 breast cancer cell lines treatment with *Bothrops jararaca* venom. The characteristics of the included studies are summarized in Table 1.

### 2.3. Analysis of Snake Venom

A total of 26 studies were reviewed to evaluate the anticancer activity of snake venom and its bioactive protein components in breast cancer. Among the snake venoms used across the studies, *Bothrops jararacussu* (*n* = 4), *Walterinnesia aegyptia* (*n* = 3), and *Naja kaouthia* (*n* = 2) were used in more than one study.

Among them, ten studies employed crude venom; one study [18] compared the antitumor effects of venom derived from four different snake species: *Bitis arietans*, *Cerastes gasperettii*, *Echis coloratus*, and *Echis pyramidum*. The remaining 16 studies investigated bioactive compounds, which were categorized into enzymatic proteins, non-enzymatic proteins, and peptide-based agents. The enzymatic protein used in the experiment was BthTX-II [36,37], an Asp-49 phospholipase A_2_. Non-enzymatic proteins included γCdcPLI [26], a phospholipase A_2_ inhibitor, and Lebecin [28], a C-type lectin-like protein. Although BthTX-I [20] and BnSP-6 [33,35] are structurally classified as Lys-49 phospholipase A_2_ isoforms, they exhibit little to no catalytic activity and are thus functionally categorized as non-enzymatic proteins. Peptide-based agents included cardiotoxins such as CTX-I [27] and CTX-III [34], recombinant cytotoxin II [23], and the synthetic cytotoxic peptide NKCT1 conjugated with gold nanoparticles [22]. Additional peptides included integrin-targeting or antimicrobial agents such as Crotalicidin, NA-CATH-ATRA-1 [25], dimeric disintegrins [31], contortrostatin utilized a liposomal formulation [40], Serine Proteinase-Associated Disintegrin-1 (SPAD-1) [21], and pBmje [43].

Nanoparticle (NP)-based delivery systems have emerged as a promising strategy to enhance the therapeutic potential of snake venom in cancer treatment. A total of six studies incorporated NPs to deliver venom or venom-derived compounds, including silica nanoparticles in four studies [19,38,39,41], chitosan NPs in one study [29], and gold NPs offer in one study [22]. NPs can enhance the accumulation of anticancer agents at tumor sites by facilitating intracellular delivery via endocytosis or phagocytosis, thereby improving therapeutic efficacy while minimizing systemic toxicity [48]. Previous studies [49,50] have demonstrated that NKCT1 conjugated gold NPs offer enhanced selectivity and sustained drug delivery to cancer cells while minimizing off-target toxicity. Bhowmik et al. (2017) [22] evaluated the anticancer efficacy of NKCT1 combined with gold NPs in MDA-MB-231 and MCF-7 cell lines. Chitosan is a biocompatible, cationic polysaccharide composed of β-linked N-acetyl-D-glucosamine and D-glucosamine units. It is primarily derived through the partial deacetylation of chitin, a natural polymer found in fungi where it occurs in association with other polysaccharides [51,52]. Jimenez-Canale et al. (2022) [29] conducted a study using chitosan NPs to encapsulate *Crotalus molossus molossus* venom. The aim was to assess the hemocompatibility of the formulation and to evaluate whether the venom retained its cytotoxic activity when applied to T-47D breast cancer cells. And Swenson et al. (2014) [40] investigated contortrostatin, a dimeric disintegrin derived from the venom of *Agkistrodon contortrix contortrix*, utilized a liposomal formulation for intravenous administration to demonstrate antitumor and antiangiogenic effects in xenograft model of human breast cancer.

The IC_50_ values, defined as the concentration of venom required to reduce cell viability by 50% after 24 h of exposure, are summarized in Table 2 and Table 3. In addition, two studies [30,39] reported LC_50_ values, representing the concentration at which the venom induces cell death in 50% of the cell population. These values vary considerably depending on the breast cancer cell line examined, as well as the specific experimental conditions and duration. Analysis of IC_50_ values in studies comparing the effects of snake venom components on MDA-MB-231 and MCF-7 breast cancer cell lines revealed cell line–specific sensitivities. *Vipera raddei kurdistanica* venom [32] and BthTX-I [20] exhibited greater cytotoxic efficacy in MCF-7 cells, indicating enhanced sensitivity in hormone receptor-positive subtypes. Conversely, compounds such as Crotalicidin, NA-CATH-ATRA-1-ATRA-1 [25], and γCdcPLI [26] demonstrated higher potency in MDA-MB-231 cells, suggesting their potential effectiveness against triple-negative breast cancer phenotypes.

### 2.4. Anticancer Activity

The anticancer activities of crude snake venom or its isolated protein components in breast cancer cells are summarized in Appendix A (see the Appendix A) and Figure 2. The most frequently used experimental assay to evaluate anticancer activity across the reviewed studies was the MTT assay. The crude snake venom and bioactive protein compound consistently demonstrated a reduction in cell viability and an increase in cytotoxicity in a dose- and time-dependent manner.

Several studies, inculding Gimenes et al. (2017) [26], Hiu et al. (2021) [27], Malekara et al. (2020) [32], and Van Petten de Vasconcelos Azevedo et al. (2019) [36], utilized lactate dehydrogenase (LDH) release assays to further confirm cytotoxicity, with elevated LDH levels indicating compromised membrane integrity and cell lysis. Al-Asmari et al. (2016) [18] reported reduced colony formation in MDA-MB-231 cells using a clonogenic survival assay, further supporting the long-term cytotoxic effects of venom treatment. Kisaki et al. (2021) [30] employed the WST-1 assay and observed that cell death in MDA-MB-231 and MCF-7 cell lines began at 2.5 μg/mL of *Bothrops jararaca* venom, with most MCF-7 cells dead at 5.0 μg/mL, while MDA-MB-231 cells showed greater resistance. Latinovi et al. (2017) [31] demonstrated that a dimeric disintegrin significantly reduced the viability of MDA-MB-231 cells at concentrations above 50 nM, as measured by the PrestoBlue ^TM^ assay.

In vivo studies have provided substantial evidence supporting the anticancer activity of snake venoms against breast cancer. Badr et al. [38] demonstrated that *Walterinnesia aegyptia* venom, particularly when conjugated with silica NPs, significantly inhibited tumor growth in BALB/c mice bearing MDA-MB-231 xenografts, with enhanced apoptotic activity confirmed through Annexin V/PI and JC-1 assays, and modulation of pro- and anti-apoptotic protein expression. Similarly, Soliman et al. [39] reported that *Naja haje* venom with silica NPs suppressed tumor progression in Wistar rats, accompanied by reduced IL-6 and TNF-α levels and increased expression of pro-apoptotic genes such as p53, BAX, and caspase-3. Swenson et al. [40] focused on the antiangiogenic potential of contortrostatin, a disintegrin from *Agkistrodon contortrix contortrix*, and demonstrated a marked reduction in neovascularization in MDA-MB-231 xenografts following intravenous administration of liposomal formulations. In a CAM model, Van Petten de Azevedo et al. [37] showed that BthTX-II, a phospholipase A_2_ from Bothrops jararacussu, effectively reduced tumor size, weight, and vascular caliber, further highlighting the role of venom-derived compounds in targeting tumor angiogenesis. Collectively, these findings underscore the multi-targeted in vivo efficacy of various snake venoms in breast cancer models, involving apoptosis induction, inflammatory suppression, and angiogenesis inhibition.

Enhancement of cytotoxic activity through NP-based delivery systems was reported in multiple studies. Al-Sadoon et al. (2012) [19] showed that *Walterinnesia aegyptia* venom exhibited an IC_50_ of 50 ng/mL in MDA-MB-231 and MCF-7 cells, whereas its combination with silica NPs reduced the IC_50_ to 20 ng/mL, indicating increased potency. In vivo validation by Badr et al. (2013) [38] using a xenograft model in BALB/c mice revealed that snake venom combined with silica NPs more effectively inhibited tumor growth than venom alone. Similarly, Jimenez-Canale et al. (2022) [29] demonstrated that chitosan NP-entrapped *Crotalus molossus molossus* venom induced a more pronounced reduction in T-47D breast cancer cell viability compared to the crude venom, highlighting the role of nanocarriers in enhancing antitumor efficacy.

### 2.5. Anticancer Mechanism

The anticancer mechanisms of crude snake venom and its isolated protein components in breast cancer cells are summarized in Appendix A (see the Appendix A). These tables summarize diverse pathways and molecular targets involved, including apoptosis induction, inhibition of metastasis, and modulation of oncogenic signaling. This compilation provides an integrated overview of the multifaceted biological activities underpinning the venom’s therapeutic potential.

#### 2.5.1. Apoptosis Induction

Apoptosis induced by snake venom has been evaluated using various flow cytometry-based assays, including the Annexin V binding assay, Annexin V/PI dual staining, Annexin V-FITC/PI staining, and TUNEL assay. Multiple studies have shown that treatment with snake venom components leads to an increased percentage of early and late apoptotic cells, as well as a rise in necrotic cell populations, indicating that snake venom induces both apoptosis and necrosis in breast cancer cells.

Several studies have investigated the pro-apoptotic effects of snake venom components through the disruption of matrix metalloproteinase (MMP). Badr et al. (2013) [38] demonstrated that treatment with *Walterinnesia aegyptia* venom in MDA-MB-231 xenograft-bearing BALB/c mice resulted in a significant decrease in MMP, as measured by JC-1 staining. Similarly, Malekara et al. (2020) [32] reported that *Vipera raddei kurdistanica* venom induced a loss of MMP in both MDA-MB-231 and MCF-7 breast cancer cells, as assessed using the JC-10 assay. Together, these findings indicate that snake venom can initiate apoptosis by compromising mitochondrial integrity, thereby triggering downstream apoptotic signaling.

Snake venom components have been shown to modulate both oxidative stress and inflammatory responses in breast cancer models. These mechanisms are closely associated with apoptosis induction. Al-Asmari et al. (2016) [18] evaluated four different snake venoms in MDA-MB-231 cells and observed a significant increase in ROS production, accompanied by a decrease in pro-inflammatory cytokines, including IL-8 and IL-6, indicating a dual role in promoting oxidative stress while reducing inflammation. Similarly, Badr et al. (2013) [38] demonstrated that *Walterinnesia aegyptia* venom induced high levels of ROS, hydroperoxides, and nitric oxide in a BALB/c mouse xenograft model bearing MDA-MB-231 tumors. These findings were further supported by Badr et al. (2014) [41] in human breast cancer tissue samples, confirming the venom’s capacity to amplify intracellular oxidative stress markers. In a more recent study, Soliman et al. (2024) [39] showed that *Naja haje* venom with silica NPs in MDA-MB-231-bearing Wistar rats led to a reduction in inflammatory cytokines, particularly IL-6 and TNF-α, suggesting that snake venom may also exert anti-inflammatory effects in vivo.

Several studies have demonstrated that snake venom components induce apoptosis in breast cancer cells by modulating both intrinsic and extrinsic apoptotic pathways. Badr et al. (2013) [38] reported that *Walterinnesia aegyptia* venom increased apoptotic activity in MDA-MB-231 xenograft-bearing BALB/c mice, as evidenced by the upregulation of caspase-3, -8, and -9, as well as pro-apoptotic proteins including Bak, Bax, and Bim, along with the downregulation of anti-apoptotic proteins such as Bcl-2, Bcl-xL, and Mcl-1. These findings were further supported by Badr et al. (2014) [41], who observed similar increases in caspase activity in human breast cancer tissue samples, suggesting the involvement of both mitochondrial and death receptor-mediated apoptosis. Consistent with these results, Bezerra et al. (2019) [20] demonstrated that BthTX-I, a Lys49-PLA_2_ from *Bothrops jararacussu*, upregulated pro-caspase-3 and -8 while downregulating Bcl-2 in MCF-7 cells, indicating activation of the caspase cascade and suppression of survival signaling. Similarly, Malekara et al. (2020) [32] reported that *Vipera raddei kurdistanica* venom induced apoptosis in MCF-7 and MDA-MB-231 cells by increasing BAX expression and decreasing BCL-2, further supporting the role of mitochondria-dependent apoptosis. In addition, Van Petten de Vasconcelos Azevedo et al. (2019) [36] showed that BthTX-II, a PLA_2_-like toxin from *Bothrops jararacussu*, promoted the expression of TNF, TNFRSF1A, CASP8, and TP53, while suppressing MDM2, a negative regulator of p53, in MDA-MB-231 cells. These molecular alterations suggest that the venom activates both extrinsic apoptosis via death receptors and p53-mediated mitochondrial apoptosis. Gimenes et al. (2017) [26] investigated the effects of γCdcPLI, a phospholipase A_2_ inhibitor derived from *Crotalus durissus collilineatus*, and found no evidence of caspase-mediated apoptosis in MDA-MB-231 cells. Their study reported upregulation of TNF, but downregulation of several pro-apoptotic and death receptor-associated genes, including BAD, BAX, BCL2, BCL2L1, TNFRSF10B, TNFRSF1A, and CASP8. Notably, no activation of caspase-3 or -7 was observed. At the protein level, although p-p53 and p-ERK levels were elevated, the PI3K/Akt signaling pathway was inhibited, as evidenced by decreased levels of phosphorylated Akt. These findings suggest that γCdcPLI does not induce apoptosis through classical caspase pathways, but may instead influence alternative stress-related or non-apoptotic mechanisms. Collectively, these studies indicate that snake venom exerts pro-apoptotic effects in breast cancer cells through the activation of both intrinsic and extrinsic apoptotic pathways, involving caspase activation, modulation of Bcl-2 family proteins, and p53-related signaling mechanisms.

#### 2.5.2. Inhibition of Cell Adhesion, Migration, Invasion, and Proliferation

Numerous studies have investigated the anti-metastatic potential of snake venom by evaluating its effects on cell adhesion, migration, and invasion, which are critical steps in the metastatic cascade of breast cancer. These studies employed established experimental methods, including adhesion assays, wound healing assays, Boyden chamber assays, and Matrigel-based transwell invasion assays. The results consistently showed that treatment with snake venom or venom-derived peptides reduced cell adhesion by decreasing attachment to extracellular matrix substrates, and also suppressed cell motility, migration, and invasion. The anti-proliferative effects of snake venom components have been evaluated in numerous studies using various assays, including the carboxyfluorescein succinimidyl ester (CFSE) dilution assay and conventional cell proliferation assays. Badr et al. (2013) [38] reported that *Walterinnesia aegyptia* venom significantly reduced IGF-1-mediated proliferation of MDA-MB-231 tumors in xenografted BALB/c mice, suggesting interference with growth factor signaling pathways that promote cell cycle progression. Similarly, Silva et al. (2018) [33] demonstrated that BnSP-6, from *Bothrops pauloensis*, significantly inhibited the proliferation of MDA-MB-231 cells, both in the presence and absence of basic fibroblast growth factor (bFGF). These results indicate that the venom exerts a direct anti-proliferative effect, independent of external mitogenic stimulation. Collectively, these findings highlight that snake venom effectively impairs multiple steps of the metastatic process in breast cancer cells by inhibiting their adhesion, migration, invasion and proliferation abilities.

Bhowmik et al. (2017) [22] investigated the antiproliferative effects of gold NP-conjugated NKCT1 (GNP-NKCT1) in both ER-positive MCF-7 and ER-negative MDA-MB-231 breast cancer cell lines. The study demonstrated that GNP-NKCT1 significantly inhibited cell proliferation in MDA-MB-231 cells only when 17β-estradiol was added to induce estrogen receptor signaling. In contrast, in MCF-7 cells, the presence of 17β-estradiol reduced the cytotoxicity of GNP-NKCT1, whereas its absence enhanced reduction in cell viability. These results suggest that GNP-NKCT1 exerts its anticancer effects through an estrogen receptor-dependent mechanism and that its efficacy may be reduced under estrogenic stimulation.

Several studies have examined the effects of snake venom on EMT, a key process in cancer metastasis. Tsai et al. (2016) [34] demonstrated that CTX-III, a cytotoxin derived from *Naja naja atra*, modulated EMT marker expression in EGF-stimulated MDA-MB-231 cells. Treatment with CTX-III led to a decrease in mesenchymal markers, including N-cadherin and vimentin, along with a reduction in epithelial adherens junction proteins. Conversely, E-cadherin expression was markedly upregulated, indicating a reversal of EMT and the restoration of epithelial characteristics. Similarly, Van Petten de Vasconcelos Azevedo et al. (2019) [36] reported that BthTX-II, a PLA_2_-like protein from *Bothrops jararacussu*, regulated EMT-associated genes in MDA-MB-231 cells by upregulating CDH1 and downregulating MCAM, CTNNB1 (β-catenin), and TWIST1, which are transcription factors and adhesion molecules associated with mesenchymal transition. These findings suggest that snake venom and its components may inhibit metastatic progression in breast cancer by modulating EMT, thereby preserving epithelial integrity and reducing mesenchymal characteristics.

#### 2.5.3. Cell Cycle Regulation

Multiple studies have demonstrated that snake venom or its components can modulate the cell cycle and induce apoptosis in breast cancer cells. In Badr et al. (2014) [41] and Bhowmik et al. (2017) [22], snake venom led to a decrease in the proportion of cells in the S phase and an accumulation in the G0/G1 phase, indicating cell cycle arrest at the G1/S checkpoint and suppression of DNA synthesis. In Gallego-Londoño et al. (2025) [25] and Van Petten de Vasconcelos Azevedo et al. (2019) [36] reported a significant increase in sub-G1 or sub-G0 populations, which is indicative of apoptotic cell death associated with DNA fragmentation. In addition, Silva et al. (2018) [33], Van Petten de Vasconcelos Azevedo et al. (2019) [36], and Van Petten de Vasconcelos Azevedo et al. (2022) [37] consistently showed cell cycle arrest at the G2/M phase, suggesting that cells were unable to proceed to mitosis after DNA replication. This G2/M arrest is often associated with DNA damage responses or intracellular stress signaling. Collectively, these findings indicate that snake venom exerts its antiproliferative effects by inducing cell cycle arrest at either the G1/S or G2/M checkpoints, accompanied by the activation of apoptosis pathways.

#### 2.5.4. Membrane Disruption Induction

Gallego-Londoño et al. (2025) [25] analyzed cell membrane integrity through confocal microscopy, demonstrating that snake venom led to enhanced membrane permeability and subsequent cell death, particularly at higher concentrations. Further fluorescence intensity assays revealed that while the MMP remained intact, venom exposure caused membrane leakage and cell death independent of classical apoptosis pathways, suggesting that cytotoxicity was primarily due to the loss of plasma membrane integrity rather than intrinsic apoptotic signaling. Similarly, Hiu et al. (2021) [27] employed Calcein-AM/PI double staining and observed a marked increase in membrane permeabilization following treatment with high concentrations of CTX-I over a short period. These findings support the conclusion that snake venom disrupts plasma membrane integrity, leading to uncontrolled leakage of cellular contents, ionic imbalance, and ultimately, cell death.

#### 2.5.5. Antiangiogenic Induction

The antiangiogenic potential of snake venom and its protein components has been demonstrated through various in vitro, ex vivo, and in vivo models. In an ex vivo aortic ring assay, Gimenes et al. (2017) [26] reported a significant reduction in the number of sprouting elongated vessels derived from aortic fragments of BALB/c mice. In vitro experiments using HUVECs revealed that venom exposure inhibited bFGF-induced vessel formation and was associated with a decrease in vascular endothelial growth factor (VEGF) expression. Similarly, Van Petten de Vasconcelos Azevedo et al. (2022) [37] demonstrated that snake venom inhibited angiogenesis in HUVECs. Their ex vivo and in vivo experiments corroborated these findings, showing a reduction in microvessel formation from aortic fragments and a decreased angiogenesis rate in BALB/c mice. Furthermore, CAM assays revealed a reduction in vessel caliber, as well as decreased tumor size and weight, indicating systemic antiangiogenic and antitumor effects. In a xenograft model, Swenson et al. (2004) [40] investigated the effects of contortrostatin, a disintegrin derived from *Agkistrodon contortrix contortrix*, and observed a marked suppression of tumor-induced angiogenesis in MDA-MB-231-bearing mice. CD31 immunostaining and quantification of vascular hotspots demonstrated an 82% reduction in neovascularization following intratumoral injection and a 94% reduction following intravenous administration of liposomal contortrostatin, highlighting the therapeutic potential of venom-derived peptides in targeting tumor vasculature. Collectively, these findings indicate that snake venom exerts potent antiangiogenic effects by inhibiting endothelial cell vessel formation, suppressing VEGF expression, and reducing neovascularization in tumor models, suggesting its promise as an antiangiogenic agent for breast cancer therapy.

#### 2.5.6. Molecular Mechanisms of Snake Venom-Induced Anticancer Activity

Multiple studies have shown that snake venom components interfere with key oncogenic signaling pathways in breast cancer cells. Bhowmik et al. (2017) [22] reported that GNP-NKCT1, derived from *Naja kaouthia*, suppressed mitogenic signaling by inhibiting cyclin D1–CDK4 and inactivating the MAPK pathway via reduced phosphorylation of ERK1/2 and p38, alongside downregulation of PI3K/Akt and NF-κB signaling in MCF-7 cells. Similarly, Gimenes et al. (2017) [26] demonstrated that γCdcPLI, a phospholipase A_2_ inhibitor from *Crotalus durissus collilineatus*, elevated p-p53 and p-ERK levels while inhibiting PI3K/Akt pathway activity, as indicated by reduced phosphorylated Akt in MDA-MB-231 cells. Furthermore, Tsai et al. (2016) [34] showed that CTX-III from *Naja naja atra* significantly suppressed NF-κB activity, inhibited EGFR phosphorylation, and attenuated downstream PI3K/Akt and ERK1/2 activation, in addition to repressing MMP-9 expression in EGF-stimulated MDA-MB-231 cells. Collectively, these findings suggest that snake venom can effectively block proliferative and survival signaling cascades, contributing to tumor progression inhibition.

Other studies have highlighted the ability of snake venom components to modulate cell cycle progression by altering the expression of key regulatory proteins. Bhowmik et al. (2017) [22] further demonstrated that GNP-NKCT1 inhibited cyclin D1–CDK4 complexes, leading to G1 phase arrest in MCF-7 cells. Silva et al. (2018) [33] reported that BnSP-6, from *Bothrops pauloensis*, downregulated cell cycle regulators including CCND1, CCNE1, CDC25A, CHEK2, E2F1, and NF-κB, while upregulating CDH1 in MDA-MB-231 cells. Likewise, Van Petten de Vasconcelos Azevedo et al. (2019) [36] showed that BthTX-II, from *Bothrops jararacussu*, suppressed the expression of CCND1, CCNE1, CDC25A, E2F1, AKT1, and AKT3, while upregulating ATM, a key DNA damage response gene. These findings indicate that venom-derived peptides exert anti-proliferative effects by modulating the cell cycle machinery at both transcriptional and post-translational levels.

#### 2.5.7. Enhanced Antitumor Efficacy of Snake Venom via NP-Based Delivery

Several studies have demonstrated that snake venom-loaded NPs significantly enhance therapeutic efficacy against breast cancer cells. Badr et al. (2013) [38] reported that *Walterinnesia aegyptia* venom loaded onto silica NPs more effectively inhibited tumor growth in MDA-MB-231 xenograft-bearing BALB/c mice compared to venom alone. In a follow-up study using human breast cancer tissue samples, Badr et al. (2014) [41] showed that the combination of *Walterinnesia aegyptia* loaded silica NPs markedly enhanced cell growth arrest and apoptosis by amplifying intracellular oxidative stress, as evidenced by increased levels of ROS, hydroperoxides, and nitric oxide. Similarly, Jimenez-Canale et al. (2022) [29] found that *Crotalus molossus molossus* venom encapsulated in chitosan NPs significantly reduced cell viability and induced morphological alterations in T-47D breast cancer cells, highlighting chitosan’s potential as a biocompatible carrier. In another xenograft model, Soliman et al. (2024) [39] demonstrated that co-administration of *Naja haje* venom with silica NPs in albino Wistar rats bearing MDA-MB-231 tumors led to improved inflammatory cytokine profiles, modulation of tumor marker levels, and upregulation of tumor suppressor genes. Swenson et al. (2004) [40] evaluated the intravenous delivery of liposomal contortrostatin, a disintegrin derived from *Agkistrodon contortrix contortrix*, in an MDA-MB-231 xenograft model. The liposomal formulation exhibited prolonged circulation time and passive tumor accumulation without inducing interactions with platelets or immune responses, thereby improving both efficacy and systemic safety compared to the free form of the venom. Collectively, these findings suggest that nanocarrier-based delivery systems substantially enhance the bioavailability, tumor targeting, and therapeutic index of snake venom components, and offer a promising strategy for overcoming delivery-related limitations in venom-based breast cancer therapy.

**Figure 2 toxins-17-00477-f002:**
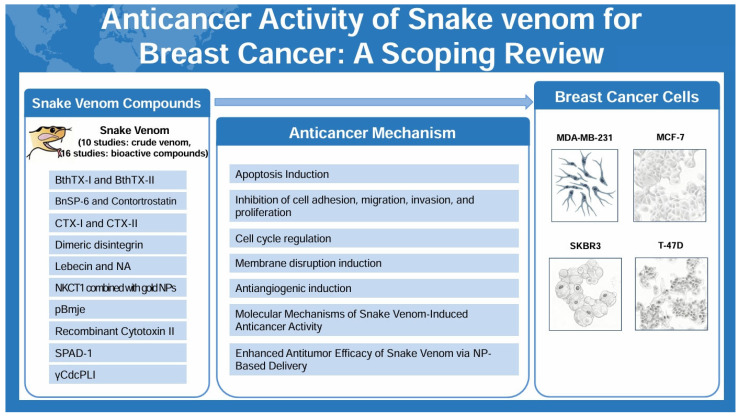
Anticancer activities of snake venom compounds against breast cancer cells. This figure illustrates the anticancer mechanisms of snake venom delivered via nanocarriers, emphasizing induction of cell cycle arrest, apoptosis, inhibition of angiogenesis, and modulation of epithelial–mesenchymal transition to effectively target cancer cells.

## 3. Discussion

### 3.1. Main Finding and Its Implication

This scoping review draws together findings from 26 experimental studies investigating the anticancer effects of snake venom and its bioactive proteins or peptides in breast cancer models. The evidence spans in vitro, ex vivo, and in vivo systems. Across these studies, both crude venoms and purified components—such as phospholipases A_2_, L-amino acid oxidases, disintegrins, and cytotoxins—were shown to exert marked cytotoxic activity against a range of breast cancer cell lines, including MDA-MB-231 (triple-negative), MCF-7 (ER/PR-positive), and SK-BR-3 (HER2-positive). The reported mechanisms are varied. Apoptosis was a common outcome, occurring through mitochondrial depolarization and caspase activation via both intrinsic and extrinsic pathways. Several studies also noted elevated levels of intracellular ROS, leading to oxidative stress and cell cycle arrest. Additional effects included loss of membrane integrity—evident from LDH release and morphological alterations—and strong suppression of cell migration, invasion, and EMT, particularly in aggressive phenotypes. Anti-angiogenic activity was observed in multiple models, characterized by reduced VEGF expression, inhibition of tube formation in HUVEC assays, and disruption of integrin-mediated adhesion. Together, these findings suggest that snake venom-derived agents have the capacity to disrupt multiple hallmarks of breast cancer progression, including growth, invasion, and angiogenesis.

Notably, several studies highlight that the sensitivity to snake venom components varies according to breast cancer subtype. For instance, venoms such as that from Vipera raddei kurdistanica and the protein BthTX-I demonstrated greater efficacy against ER-positive MCF-7 cells, whereas compounds like crotalicidin and γCdcPLI displayed pronounced cytotoxicity towards triple-negative models exemplified by the MDA-MB-231 cell line. These subtype-dependent responses point to the potential of tailoring snake venom-derived agents within precision oncology frameworks [53]. Advances in nanoparticle-based delivery—including systems utilizing silica, gold, chitosan, and liposomal carriers—have been shown to increase tumor selectivity and cellular uptake, as well as overall antitumor activity. Importantly, these technologies also help limit off-target toxicity [54]. Evidence from animal studies, including xenograft mouse models and the chick chorioallantoic membrane assay, further supports both the antitumor and anti-angiogenic properties of these agents, especially when delivered via nanocarriers or in combination with standard chemotherapeutics. These preclinical data consistently demonstrate greater selectivity for malignant cells over normal tissue [55,56]. Collectively, current research positions snake venom–based compounds as promising multi-targeted anticancer agents that may help address pharmacoresistance and mitigate the side effects associated with conventional therapies [17].

### 3.2. Study Strength and Limitation

This review’s main strength lies in its broad synthesis of studies investigating the anticancer properties of snake venom and its bioactive constituents in breast cancer models, incorporating evidence from in vitro, ex vivo, and in vivo experimental systems. By encompassing research on a wide array of snake species and venom fractions—from crude venoms to isolated proteins and peptides—this review highlights the diversity of underlying mechanisms and acknowledges the therapeutic opportunities these agents may present [57]. Nonetheless, several limitations should be considered when interpreting the findings. A substantial proportion of the available evidence is derived from in vitro studies, with relatively few in vivo validations, making it difficult to predict whether these results will translate to clinical contexts. There are also several significant challenges that limit the translation of snake venom-derived compounds into clinically viable breast cancer therapeutics [5]. A major obstacle lies in the inherent toxicity of snake venom, necessitating precise dosage optimization to maximize anticancer efficacy while minimizing adverse effects [58]. Isolating and standardizing bioactive components from complex venom mixtures remain technically demanding, resulting in variability across studies that hinders reproducibility and comparability [59]. Furthermore, the scarcity of rigorous in vivo and clinical trial data creates a translational gap, compounded by ethical and safety considerations inherent to venom-based therapeutics. Overcoming these barriers requires advancements in purification technologies, nanoparticle-based delivery systems, and comprehensive safety profiling, alongside carefully designed clinical studies [60]. Addressing these limitations is crucial to harness the full potential of snake venoms as novel anticancer agents.

### 3.3. Future Perspective

Ongoing research should emphasize the discovery and molecular characterization of new bioactive compounds from a wide range of snake species. Employing advanced analytical methods such as venomics and proteomics will help address the current limitations of low yield and restricted diversity among venom-derived molecules [59]. A key focus is the identification of proteins and peptides with selective activity against malignant cells, which may guide the development of more effective and targeted anticancer agents [58]. Improving drug delivery remains another critical challenge. Recent advances suggest that integrating snake venom components with carriers such as metal nanoparticles, liposomes, or extracellular vesicles can promote greater tumor selectivity, prolong drug stability, and enable controlled release, while mitigating systemic toxicity in preclinical models [61]. To realize these preclinical benefits in a clinical setting, further progress in pharmaceutical formulation is needed. In particular, new delivery technologies must be developed to enhance the stability, solubility, and tissue specificity of snake venom-derived compounds, with the ultimate goal of supporting safe and effective translation to clinical trials [62].

## 4. Conclusions

This scoping review examined the anticancer activity of a range of snake venoms and their bioactive constituents in breast cancer, drawing on evidence from in vitro, ex vivo, and in vivo models. Reported mechanisms included apoptosis induction, ROS generation, disruption of cell membrane integrity, inhibition of cell proliferation and metastasis, and regulation of tumor-associated signaling pathways such as PI3K/Akt, MAPK, and NF-κB. Several animal studies also indicated that combining certain venom-derived compounds with nanotechnology-based delivery systems or standard chemotherapeutic agents could further enhance tumor suppression. Nonetheless, most available studies were heavily weighted toward in vitro experiments, and substantial heterogeneity—stemming from differences in venom source, purification procedures, dosage, exposure time, and the cancer cell lines used—makes it difficult to generalize the findings to clinical contexts. Critical gaps also remain in the literature regarding long-term toxicity, immunogenicity, off-target effects, pharmacokinetic interactions, and the standardization of formulations. Future work should aim to identify new compounds using venomics- and proteomics-driven screening, refine formulations through advanced delivery platforms such as nanoparticles, liposomes, or extracellular vesicles, and conduct rigorous safety and efficacy testing to help close the gap between preclinical results and clinical application.

## 5. Materials and Methods

### 5.1. Study Design and Registration

This scoping review was conducted in accordance with the PRISMA-ScR guidelines and established best practices for literature synthesis [63,64] (see the Appendix A). The study protocol was preregistered with the Open Science Framework (OSF) (https://doi.org/10.17605/OSF.IO/Y3EB9, accessed on 10 August 2025).

### 5.2. Data Sources and Searches

Comprehensive literature searches were conducted across multiple electronic databases, including PubMed, EMBASE, the Cochrane Central Register of Controlled Trials, CINAHL Plus, ScienceON, the Korean Traditional Knowledge Portal, Korea Citation Index, Research Information Sharing Service, OASIS, and the Korean Medical Database. Studies published up to June 2025 were eligible for inclusion. The search terms used were ((“breast neoplasms” OR “breast cancer” OR “mammary carcinoma” OR “breast tumor” OR “breast malignancy”) AND (“snake venom” OR “snake venom” OR “viper venom” OR “elapid venom” OR “cobra venom” OR “serpent venom” OR “disintegrin” OR “snake toxin”) AND (“in vivo” OR “in vitro” OR “experimental study” OR “cell line” OR “mouse model” OR “animal model”)). The search terms for each database are included in Appendix A.

### 5.3. Study Selection

Two reviewers independently screened all retrieved titles and abstracts to identify studies investigating the anticancer effects of snake venom or venom-derived peptides/proteins in breast cancer models. Studies were included if they met the following criteria: (1) employed in vitro, ex vivo, or in vivo breast cancer models; (2) assessed cytotoxic, anti-proliferative, anti-migratory, anti-angiogenic, or related effects of crude snake venom or purified components; and (3) reported original research data. Exclusion criteria were as follows: non-original publications (e.g., reviews, letters), studies unrelated to breast cancer, and those involving non-snake sources. Full texts of potentially eligible studies were retrieved and reviewed in detail. Any discrepancies between reviewers were resolved through discussion or, if necessary, by consultation with a third reviewer.

### 5.4. Data Extraction

Key study characteristics were independently extracted by two reviewers using a predefined template. Extracted variables included author, year of publication, country, snake species and venom fraction investigated, model type (cell line or animal), mechanistic assays performed, principal molecular or phenotypic outcomes (e.g., apoptosis, proliferation, migration, angiogenesis), delivery method, and quantitative results such as IC_50_ or LC_50_ values. When data were unclear, study authors were contacted whenever feasible. All extracted data were cross-verified for consistency, and any discrepancies were resolved through consensus. If uncertainties or missing data were identified, attempts were made to contact study investigators for clarification or additional information. In accordance with standard scoping review methodology, no formal assessment of methodological quality or risk of bias was performed.

## Figures and Tables

**Figure 1 toxins-17-00477-f001:**
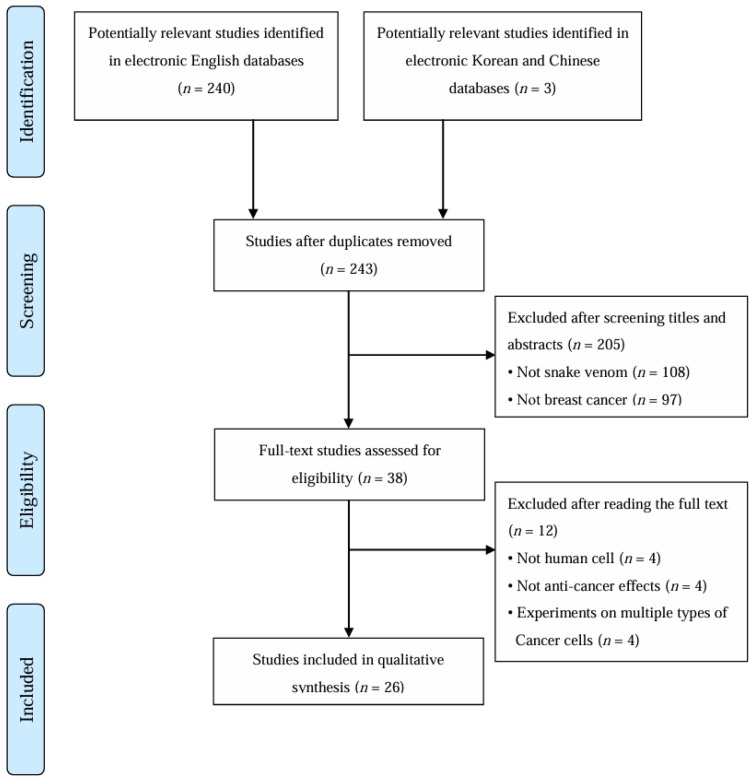
Flowchart of study selection process. The flow diagram illustrates the identification, screening, eligibility assessment, and inclusion of studies investigating snake venom effects on breast cancer.

**Table 1 toxins-17-00477-t001:** Characteristics of included studies. This table summarizes the 26 experimental studies included in the review, detailing authors, snake species and venom types tested, breast cancer cell lines and animal models used, primary mechanisms investigated, and key main findings explaining the anticancer effects observed.

Author (Year) Countries	Snake Venom	Target Cell, Animal Model	Mechanism	Main Results
Al-Asmari et al. (2016) [18] Saudi Arabia	*Bitis arietans* *Cerastes gasperettii* *Echis coloratus* *Echis pyramidum*	MDA-MB-231	Apoptotic effects by increasing the ROS Anti-proliferative effects	- reduced cell motility, colony formation, and cell invasion - increased oxidative stress - decreased the expression of pro-inflammatory cytokines and signaling proteins
Al-Sadoon et al. (2012) [19] Saudi Arabia	*Walterinnesia aegyptia*	MDA-MB-231 MCF-7	Apoptotic effects Induction of growth arrest	- induced apoptosis in MDA-MB-231 and MCF-7 cells - decreased the expression of BCL2 and enhanced the activation of caspase-3 - reduced actin polymerization and cytoskeletal rearrangement
Bezerra et al. (2019) [20] Brazil	*Bothrops jararacussu*	MDA-MB-231 MCF-7 SK-BR-3	Apoptotic effects Autophagy effects Reduction in cancer stem cells subpopulation	- apoptosis in MCF-7, SK-BR-3, and MDA-MB-231 in a dose-dependent manner by the increasing number of hypodiploid nuclei - upregulation of pro-caspase-3, -8 and Beclin-1 proteins - Induced changes in the staining profile of cancer stem cells in MDA-MB-231 cells through upregulation of CD24 receptor expression
Bhattacharya et al. (2023) [21] India	*Russell’s viper*	MCF-7	Cytotoxic effects Inhibition of adhesion	- caused visible morphological changes in MCF7 cells, including reduced cell-to-cell adhesion, cell rounding, and cell death - decreased the invasive potency of MCF7 cells - increased cell detachment from poly-L-lysine-, laminin-, and fibronectin-coated culture plate matrices, confirming disintegrin activity
Bhowmik et al. (2017) [22] India	*Naja kaouthia*	MDA-MB-231 MCF-7	Apoptotic effects Anti-metastatic effects in MCF7 cells through estrogen receptor-mediated cell cycle arrest via MAPK pathway inhibition	- increased early and late apoptosis - inactivation of CDK4, PI3K/Akt, ERK1/2, and p38 MAPK signaling pathways through inhibition of NF-κB and reduction in MMP9 activity - reduced proliferation through the estrogen receptor pathway
Derakhshani et al. (2020) [23] Iran	*Naja naja oxiana*	MCF-7	Cytotoxic effects Apoptotic effects Anti-proliferative effects Induction of cell cycle arrest	- enhanced apoptosis through the intrinsic and extrinsic pathways - increased sub-G1 phase accumulation and reduced S phase cell population - decrease the expression of MMP-3 and -9
Erlista et al. (2023) [24] Indonesia	*Naja kaouthia*	MCF-7	Characterize and identify peptides from the snake venom of *Naja kaouthia* as anticancer	- The 25% methanol peptide fraction showed potent anticancer activity against MCF-7 cells, with a high selectivity index of 12.87. - WWSDHR and IWDTIEK showed strong potential as breast cancer therapeutics by binding to the active site of EGFR.
Gallego-Londoño et al. (2025) [25] Colombia	*Crotalus durissus* *Naja atra*	MDA-MB-231 MCF-7	Cytotoxic effects Induction of membrane disruption effects	- reduced cell viability - induced membrane permeabilization and membrane disruption - cleaved caspase-9 or PARP were not detected
Gimenes et al. (2017) [26] Brazil	*Crotalus durissus collilineatus*	MDA-MB-231 MCF-7 HUVEC	Cytotoxic effects Anti-metastatic effects Anti-angiogenic effects Anti-tumoral effects via PI3K/Akt pathway	- cytotoxic to breast cancer cells (MDA-MB-231 cells >MCF-7 cells) - mediated apoptosis pathways such as p53, MAPK-ERK, BIRC5 and MDM2 - decreased MDA-MB-231 cells adhesion, migration and invasion - reduced the production of VEGF - decreased PGE2 levels in MDA-MB-231 cells - inhibited gene and protein expression of the PI3K/Akt pathway - inhibited endothelial cell adhesion and migration, and suppressed angiogenesis by blocking tube formation in HUVECs
		BALB/c mice (6 wk) Aortic fragments	Anti-angiogenic effects	- reduced the number of sprouting elongated vessels
Hiu et al. (2021) [27] Malaysia	*Naja sumatrana*	MCF-7	Necroptosis effects Induction of membrane permeabilization and loss of membrane integrity	- induced the loss of membrane integrity in a concentration dependent manner - high CTX-I concentration (at >29.8 μg/mL) induced necroptosis - The cell death pattern of MCF-7 cells shifted from apoptosis to necroptosis with increasing concentrations of CTX-I.
Jebali et al. (2014) [28] Tunisia	*Macrovipera lebetina*	MDA-MB-231	Anti-proliferative effects Inhibition of adhesion and migration	- inhibited MDA-MB-231 cells proliferation - inhibited the integrin-mediated attachment of these cells to different adhesion substrata in a dose-dependent manner - blocked MDA-MB-231 cells migration towards fibronectin and fibrinogen
Jimenez-Canale et al. (2022) [29] Mexico	*Crotalus molossus molossus*	T-47D	Cytotoxic effects	- *Crotalus molossus molossus* combined with chitosan NPs inhibited viability and induced morphological changes in the T-47D cells.
Kisaki et al. (2021) [30] Brazil	*Bothrops Jararaca*	MDA-MB-231 MCF-7	Describe the quantitative changes in proteomics of MCF7 and MDA-MB-231 cell lines treatment with Bothrops jararaca snake venom	- Sub-toxic doses (0.63 μg/mL or 2.5 μg/mL) of Bothrops jararaca venom demonstrated potential to modulate cancer-related protein targets involved in cell growth and invasion, including histone H3, SNX3, HEL-S-156an, MTCH2, RPS, MCC2, IGF2BP1, and GSTM3.
Latinovi et al. (2017) [31] Slovenia	*Vipera ammodytes ammodytes*	MDA-MB-231	Inhibition of migration Anti-metastatic effects	- reduced the cell viability in MDA-MB-231 cells - inhibited MDA-MB-231 cells migration
Malekara et al. (2020) [32] Iran	*Vipera raddei kurdistanica*	MDA-MB-231 MCF-7	Cytotoxic effects and Anti-proliferation effects via ROS mediated apoptosis	- reduced the cell viability in a time- and dose-dependent manner - increased apoptosis
Silva et al. (2018) [33] Brazil	*Bothrops pauloensis*	MDA-MB-231 MCF-7	Cytotoxic effects Genotoxic effects Anti-proliferative effects Induction of cell cycle arrest	- inhibited MDA-MB-231 cells proliferation - increased the percentage of TUNEL-positive cells - decreased 2N (G1) and increased the G2/M phase - modulation of expression of progression cell cycle-associated genes (CCND1, CCNE1, CDC25A, CHEK2, E2F1, CDH-1, NF-kB)
Tsai et al. (2016) [34] Taiwan	*Naja naja atra*	MDA-MB-231	Inhibition of EGF/EGFR-mediated EMT and invasion	- induced morphological changes from elongated and spindle shape to rounded and epithelial-like shape - upregulation of E-cadherin and concurrent downregulation of N-cadherin and Vimentin protein levels - decreased the expression of Snail and Twist in EGF-induced MDA-MB-231 cells - inhibited the EGFR phosphorylation and downstream activation of PI3K/Akt and ERK1/2 - suppressed EGF/EGFR-mediated EMT and invasion of MDA-MB-231 cells
Van Petten de Vasconcelos Azevedo et al. (2016) [35] Brazil	*Bothrops jararacussu*	MDA-MB-231	Cytotoxic effects Apoptosis effects Autophagy effects Inhibition of adhesion and migration Anti-angiogenic effects Anti-metastatic effects	- stimulated the autophagy process as evidenced by labeling of autophagic vacuoles - induced both early and late apoptosis - upregulation of different genes related to the apoptosis pathway (TNF, TNFRSF10B, TNFRSF1A and CASP8) - decreased expression of anti-apoptotic genes (BCL2 and BCL2L) - increased gene expression of BRCA2 and TP53 - downregulation of Angiopoetin 1 gene
Van Petten de Vasconcelos Azevedo et al. (2019) [36] Brazil	*Bothrops jararacussu*	MDA-MB-231	Apoptosis effects Autophagy effects Induction of cell cycle arrest Anti-metastatic effects	- caused cell death of MDA-MB-231 cells by inducing apoptosis and autophagy in a dose-dependent manner - decreased the proliferation and inhibited cell cycle progression - upregulation of the ATM gene (CCND1, CCNE1, CDC25A, E2F1, AKT1 and AKT3) - inhibited the EMT by increasing E-cadherin (CDH-1) - decreased TWIST1, CTNNB1, vimentin and cytokeratin-5
Van Petten de Azevedo et al. (2022) [37] Brazil	*Bothrops jararacussu*	MDA-MB-231 HUVEC	Anti-angiogenic effects	- inhibited cell adhesion, proliferation, and migration of HUVECs - reduction in the levels of EGF - inhibited the migration and proliferation of HUVECs in co-culture with MDA-MB-231 cells
		Chick embryos (3 d) MDA-MB-231		- reduction in vessel caliber, as well as in tumor size and weight
		BALB/c mice (6 wk) Aortic fragments		- inhibited the sprouting angiogenic process
Badr et al. (2013) [38] Saudi Arabia	*Walterinnesia aegyptia*	[Xenograft] BALB/c mice (10 wk, 22–25 g) MDA-MB-231	Cytotoxic effects Apoptotic effects Anti-proliferative effects	- reduced breast tumor volumes - increased ROS, hydroperoxides, and nitric oxide - reductions in the levels of chemokines CXCL9, CXCL10, CXCL12, CXCL13, and CXCL16 - decreased surface expression of cognate chemokine receptors CXCR3, CXCR4, CXCR5, and CXCR6 - inhibition of IGF-1-induced proliferation of MDA-MB-231 cells - enhanced caspase-3, -8, and -9 activity - induced disruption of mitochondrial membrane potential - decreased phosphorylation of key signaling proteins: AKT, ERK, and IκBα - downregulation of cyclin D1, survivin, and anti-apoptotic Bcl-2 family proteins (Bcl-2, Bcl-xL, and Mcl-1) - upregulation of cyclin B1 and pro-apoptotic Bcl-2 family members (Bak, Bax, and Bim)
Soliman et al. (2024) [39] Egypt	*Naja haje*	[Xenograft] Albino Wistar rats (7–9 wk, 100–120 g) MDA-MB-231	Anti-cancer efficacy	- *Naja haje* combined with silica NPs improved inflammatory cytokine and tumor marker profiles, increased the expression of tumor-suppressor genes, and enhanced apoptosis and necrosis.
Swenson et al. (2004) [40] United States	*Agkistrodon contortrix contortrix*	[Xenograft] Mice (5 wk) MDA-MB-231	Anti-angiogenic effects Anti-proliferative effects	- Intravenous liposomal contortrostatin prolonged circulatory half-life, passively accumulated in the tumor, exhibited no platelet reactivity, and was not recognized by the immune system
Badr et al. (2014) [41] Egypt	*Walterinnesia aegyptia*	Human Breast cancer tissue samples	Anti-proliferative effects	- inhibited breast cancer cell proliferation by inducing cell cycle arrest and promoting apoptosis - increased the activities of caspase-3, -8 and -9 - increased ROS, hydroperoxides, and nitric oxide
Jokhio et al. (2005) [42] Pakistan	*Cobra*	Human Breast cancerous tissues	Anti-proliferative effects	- inhibited the formation of nucleic acids - (maximum effect was observed at 25 μg/mL)
Peña-Carrillo et al. (2021) [43] Ecuador	*Bothrops marajoensis*	MCF-7	Cytotoxic effects	- pBmje showed moderate cytotoxicity against MCF–7

h—hours; d—days; wk—weeks; BCL2—B-cell lymphoma 2; BICR5—baculoviral IAP repeat-containing protein 5; BRCA2—breast cancer type 2 susceptibility protein; CASP—caspase; CCND1—cyclin D1; CCNE1—cyclin E1; CD24—cluster of differentiation 24; CDH1—cadherin 1; CDK4—cyclin-dependent kinase 4; CDC25A—cell division cycle 25A; CHEK2—checkpoint kinase 2; CTX-I—cardiotoxin I; E2F1—E2F transcription factor 1; EGF—epidermal growth factor; EGFR—epidermal growth factor receptor; EMT—epithelial to mesenchymal transition; ERK1/2—extracellular signal-regulated kinase 1/2; GSTM3—glutathione S-transferase mu 3; H3—histone H3; HEL-S-156an—hypothetical protein HEL-S-156an; HUVEC—human umbilical vein endothelial cell; IκBα—inhibitor of nuclear factor kappa B alpha; IGF2BP1—insulin-like growth factor 2 mRNA-binding protein 1; MAPK —mitogen-activated protein kinase; MCC2—mutated in colorectal cancers 2; MDM2—mouse double minute 2 homolog; Mcl-1—myeloid cell leukemia-1; MMP—matrix metalloproteinase; MTCH2—mitochondrial carrier homolog 2; NF-κB—nuclear factor kappa B; NP—nanoparticle; PARP—poly (ADP-ribose) polymerase; PGE2—prostaglandin E2; PI3K/Akt—phosphatidylinositol 3-kinase/protein kinase B; RPS—ribosomal protein S; ROS—reactive oxygen species; SNX3—sorting nexin 3; TP53—tumor protein p53; TNF—tumor necrosis factor; TNFRSF10B—tumor necrosis factor receptor superfamily member 10B; TNFRSF1A—tumor necrosis factor receptor superfamily member 1A; TUNEL—terminal deoxynucleotidyl transferase dUTP nick end labeling; VEGF—vascular endothelial growth factor.

**Table 2 toxins-17-00477-t002:** Characteristics of included crude snake venom. This table presents inhibitory and lethal concentrations (IC_50_ and LC_50_) from studies using crude snake venoms on various breast cancer cell lines and in vivo models, illustrating venom potency and comparative sensitivity across cancer subtypes and experimental conditions.

Author (Year)	Species	Target Cell, Animal Model	IC_50_ [LC_50_]
Al-Asmari et al. (2016) [18]	*Bitis arietans* *Cerastes gasperettii* *Echis coloratus* *Echis pyramidum*	MDA-MB-231	NR
Al-Sadoon et al. (2012) [19]	*Walterinnesia aegyptia*	MDA-MB-231 MCF-7	50 ng/mL (12 h)
	*Walterinnesia aegyptia* combined with silica NPs		20 ng/mL (12 h)
Badr et al. (2013) [38]	*Walterinnesia aegyptia*	[Xenograft] BALB/c mice (10 wk, 22–25 g) MDA-MB-231	NR
Badr et al. (2014) [41]	*Walterinnesia aegyptia*	Human Breast cancer tissue samples	50 ng/mL (12 h)
	*Walterinnesia aegyptia* combined with silica NPs		20 ng/mL (12 h)
Erlista et al. (2023) [24]	*Naja kaouthia*	MCF-7	4.17 μg/mL (25% methanol peptide fraction)
Jimenez-Canale et al. (2022) [29]	*Crotalus molossus molossus*	T-47D	15.45 μg/mL
Jokhio et al. (2005) [42]	*Cobra*	Human Breast cancerous tissues	NR
Kisaki et al. (2021) [30]	*Bothrops Jararaca*	MDA-MB-231	[4.76 μg/mL]
		MCF-7	[4.50 μg/mL]
Malekara et al. (2020) [32]	*Vipera raddei kurdistanica*	MDA-MB-231	20.29 μg/mL (24 h) 11.01 μg/mL (48 h) 5.99 μg/mL (72 h) 1.27 μg/mL (96 h)
		MCF-7	18.53 μg/mL (24 h) 8.96 μg/mL (48 h) 2.14 μg/mL (72 h) 0.98 μg/mL (96 h)
Soliman et al. (2024) [39]	*Naja haje*	[Xenograft] Albino Wistar rats (7–9 wk, 100–120 g) MDA-MB-231	[0.568 mg/kg]

NR—not reported; h—hours; wk—weeks; IC—Inhibitory Concentration; LC—Lethal Concentration; NP—nanoparticle.

**Table 3 toxins-17-00477-t003:** Characteristics of included snake venom bioactive protein. This table details the specific venom-derived proteins and peptides tested for anticancer activity, including source species, target cancer cell lines, main compounds investigated, and their cytotoxic potency (IC_50_ values), highlighting compounds with potential therapeutic relevance.

Author (Year)	Species	Main Compound	Target cell	IC_50_
Bezerra et al. (2019) [20]	*Bothrops jararacussu*	BthTX-I	MDA-MB-231	>409 ± 5.34 μg/mL
			MCF-7	104.35 ± 13.21 μg/mL
			SKBR3	81.20 ± 8.58 μg/mL
Bhattacharya et al. (2023) [21]	*Russell’s viper*	SPAD-1	MCF-7	0.41 μM (24 h) 0.21 μM (48 h)
Bhowmik et al. (2017) [22]	*Naja kaouthia*	NKCT1 combined with gold NPs	MDA-MB-231 MCF-7	NR
Derakhshani et al. (2020) [23]	*Naja naja oxiana*	Recombinant Cytotoxin II	MCF-7	3.66 μg/mL
Gallego-Londoño et al. (2025) [25]	*Crotalus durissus*	Crotalicidin	MDA-MB-231	21.3 μM/mL
			MCF-7	58.9 μM/mL
	*Naja atra*	NA	MDA-MB-231	6.4 μM/mL
			MCF-7	13.4 μM/mL
Gimenes et al. (2017) [26]	*Crotalus durissus collilineatus*	γCdcPLI	MDA-MB-231	25 ± 1.72 μM/mL
			MCF-7	28 ± 4.1 μM/mL
			HUVEC	NR
Hiu et al. (2021) [27]	*Naja sumatrana*	CTX-I	MCF-7	29.80 ± 2.3 μg/mL (4 h) 19.33 ± 0.6 μg/mL (8 h) 8.15 ± 0.1 μg/mL (16 h) 9.99 ± 1.2 μg/mL (24 h)
Jebali et al. (2014) [28]	*Macrovipera lebetina*	Lebecin	MDA-MB-231	NR
Latinovi et al. (2017) [31]	*Vipera ammodytes ammodytes*	Dimeric disintegrin	MDA-MB-231	NR
Silva et al. (2018) [33]	*Bothrops pauloensis*	BnSP-6	MDA-MB-231	52.24 μg/mL
			MCF-7	NR
Swenson et al. (2004) [40] United states	*Agkistrodon contortrix contortrix*	Contortrostatin	[Xenograft] Mice (5 wk) MDA-MB-231	NR
Tsai et al. (2016) [34]	*Naja naja atra*	CTX-III	MDA-MB-231	NR
Van Petten de Vasconcelos Azevedo et al. (2016) [35]	*Bothrops pauloensis*	BnSP-6	MDA-MB-231	NR
Van Petten de Vasconcelos Azevedo et al. (2019) [36]	*Bothrops jararacussu*	BthTX-II	MDA-MB-231	NR
Van Petten de Vasconcelos Azevedo et al. (2022) [37]	*Bothrops jararacussu*	BthTX-II	MDA-MB-231 HUVEC	NR
Peña-Carrillo et al. (2021) [43]	*Bothrops marajoensis*	pBmje	MCF-7	NR

NR—not reported; h—hour; wk—weeks; HUVEC—human umbilical vein endothelial cell; IC—Inhibitory Concentration; NA—NA-CATH-ATRA-1-ATRA-1; NP—nanoparticles; SPAD-1—Serine Proteinase Associated Disintegrin-1.

## Data Availability

The raw data supporting the conclusions of this article will be made available by the authors on request.

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
