# Peer review of "Anticancer Activity of Snake Venom Against Breast Cancer: A Scoping Review"

_toxins, 2025, doi:10.3390/toxins17100477_

Round 1

Reviewer 1 Report

Comments and Suggestions for Authors

The paper 'Anticancer Activity of Snake Venom for Breast Cancer: A Scoping Review' presents a very interesting investigation into the use of snake venom on breast cancer. The methodological approach is well structured, and the data are illustrated with the inclusion of summary tables.
However, due to the substantial body of research on the effects of venom on cancer cells in vitro, the findings from in vivo experiments, with or without the use of nano carriers, and their toxicological effects, are not sufficiently emphasised. It is therefore recommended that the authors place greater emphasis on this aspect in future publications, potentially through the use of a summary table.

Author Response

We sincerely thank the reviewer for their evaluation. Please refer to the attached file for our responses to the comments.

Reviewer 2 Report

Comments and Suggestions for Authors

The manuscript addresses an innovative and intriguing topic by reviewing the potential anticancer properties of snake venom in the context of breast cancer treatment. The subject is highly relevant and up-to-date, especially given the constant search for novel therapeutic strategies against breast cancer. The authors have followed the PRISMA guidelines, which strengthens the methodological transparency and credibility of the review.

The paper is well-structured and provides a comprehensive overview of the available literature. However, a few aspects could be further improved to enhance the overall quality and impact of the manuscript:

  1. Context of Current Breast Cancer Treatments – The review would benefit from the addition of a section describing the current therapeutic strategies for breast cancer, including distinctions between molecular subtypes (e.g., hormone receptor–positive, HER2-positive, triple-negative). Such a section would not only provide readers with essential background but also allow for a more meaningful comparison of how snake venom–derived compounds might offer advantages over existing treatments.

  2. Limitations and Research Challenges – While the authors mention the limitations of the review itself, they do not elaborate sufficiently on the specific obstacles and limitations faced in experimental studies involving snake venom. These may include issues such as toxicity, dosage optimization, challenges in isolating and standardizing bioactive components, difficulties in conducting clinical trials, and ethical considerations. A critical discussion of these points would give the reader a more balanced and realistic picture of the current state of research in this area.

Overall, the manuscript presents a novel and engaging synthesis of the literature. With the addition of a more detailed discussion on existing breast cancer therapies and the research barriers to the application of snake venom, the review would gain further depth and provide even greater value to the scientific community.

Author Response

(The authors gave the same response as above.)

Reviewer 3 Report

Comments and Suggestions for Authors

This scoping review provides a  synthesis of experimental evidence on the anticancer activity of snake venom and its bioactive components against breast cancer. Although interesting, the manuscript requires major revision. The main contribution is not clear. In addition, the search failed to include all important references.  

  1. Please use keywords different from those already included in the manuscript’s title.
  2. he abstract section lacks focus on the novelty and significance of the review. It would benefit from an introductory sentence. Additionally, the abstract should present a more critical perspective on the findings, along with a concise statement of the review’s aim, key conclusions, and future outlook.
  3. The key contributions of this review appear to overlap with those of previously published literature. The authors should clarify what novel insights or perspectives they offer beyond the already described mechanisms of action.
  4. Some claims must be supported by references. For instance, lines 38-39, 142-143
  5. lines 52-55. In vivo studies (?), but only one reference has been added.
  6. Line 56-57. To better emphasize the importance and relevance of this manuscript, the authors could have cited additional studies investigating the anticancer and other biological activities of venom components. For example: https://analyticalsciencejournals.onlinelibrary.wiley.com/doi/10.1002/jat.4544, https://www.tandfonline.com/doi/full/10.1080/17460441.2025.2465364, , https://pmc.ncbi.nlm.nih.gov/articles/PMC11422003/ and others. This would help contextualize the current work, especially given the notable lack of studies specifically focused on breast cancer.
  7. Line 72: Please add a brief description to the figure legend in addition to the title.
  8. Please avoid one-sentence paragraphs. For example, lines 125-126, 212-214
  9. lines 129-130. What was the purity degree of the proteins evaluated? How was it estimated experimentally?
  10. Table 2. Please include a brief description in addition to the title.
  11. The authors could consider incorporating figures to enhance the clarity and overall understanding of the manuscript’s content.
  12. Did the authors observe any biases in the study of venoms and toxins? Family or snake?
  13. The discussion sections need to be supported by references and improved.
  14. The authors have not balanced the manuscript with the obstacles to converting snake venom toxins into anticancer drugs. The authors must explore routes and strategies to improve the translational potential.
  15. The authors have not included important studies that have evaluated peptides designed from snake toxins with effects on cancer cell lines. Toxins are usually bigger than peptides. Peptides are gaining more attention in the development of anticancer agents. Here are some examples that can enrich the manuscript: https://www.sciencedirect.com/science/article/pii/S0045206821004181 and https://www.mdpi.com/1467-3045/44/1/4

Author Response

(The authors gave the same response as above.)

Round 2

Reviewer 2 Report

Comments and Suggestions for Authors

The authors made all suggested changes.

Author Response

We sincerely thank the reviewers for their valuable comments and suggestions. We have redrawn Figure 2 to further improve the clarity and overall understanding of the manuscript.

Reviewer 3 Report

Comments and Suggestions for Authors

The authors have addressed all comments. My remaining concern is that the literature search may not comprehensively capture all relevant studies. For example, PMC11079407 was not included; likewise, as noted in my previous review, doi:10.3390/cimb44010004 was overlooked. Although the peptides in the latter were more active in osteosarcoma cell lines, they also demonstrated toxicity in the MCF-7 cell line. Both manuscripts meet the inclusion criteria described in the methodology, and a simple search reveals this inconsistency. The authors should therefore revise the tables and undertake a more thorough, validated search to ensure comprehensive coverage of the literature.

In addition, the figure included in the manuscript must be explicitly referenced and contextualised in the main text. The figure quality is poor, and the authors should clarify that it is AI-generated, as it displays features typical of AI imagery. Furthermore, the figure legend contains inconsistencies, and it is unclear whether the focus of the figure is on the delivery systems or on the mechanisms of action of snake venoms.

Author Response

We fully understand the reviewer’s concern regarding the comprehensiveness of the literature search. As noted in our response to Comment 16, the cited studies did not meet the inclusion criteria of this scoping review and were therefore excluded. We acknowledge that, despite designing search strategies tailored to each database, there is a possibility that some potentially relevant articles may not have been captured. This limitation is inherent to the nature of a scoping review. However, to ensure transparency, we have provided our detailed search strategies in the supplementary file. We expect that future research, particularly systematic reviews, will be able to employ more refined and validated search strategies to minimize such omissions.

We have redrawn Figure 2 to further improve the clarity and overall understanding of the manuscript.